# A Contrastive Framework for Neural Text Generation

**Yixuan Su**♠ **Tian Lan**◇ **Yan Wang**◇ **Dani Yogatama**♣
**Lingpeng Kong**♡ **Nigel Collier**♠
♠Language Technology Lab, University of Cambridge
◇Tencent AI Lab ♣DeepMind
♡Department of Computer Science, The University of Hong Kong
{ys484,nhc30}@cam.ac.uk
lantiangmftby@gmail.com, yanwang.branden@gmail.com
dyogatama@deepmind.com, lpk@cs.hku.hk

## Abstract

Text generation is of great importance to many natural language processing applications. However, maximization-based decoding methods (e.g., beam search) of neural language models often lead to degenerate solutions—the generated text is unnatural and contains undesirable repetitions. Existing approaches introduce stochasticity via sampling or modify training objectives to decrease the probabilities of certain tokens (e.g., unlikelihood training). However, they often lead to solutions that lack coherence. In this work, we show that an underlying reason for model degeneration is the anisotropic distribution of token representations. We present a contrastive solution: (i) *SimCTG*, a contrastive training objective to calibrate the model's representation space, and (ii) a decoding method—*contrastive search*—to encourage diversity while maintaining coherence in the generated text. Extensive experiments and analyses on three benchmarks from two languages demonstrate that our proposed approach significantly outperforms current state-of-the-art text generation methods as evaluated by both human and automatic metrics.[1]

## 1 Introduction

Open-ended neural text generation [19, 23] with Transformer [25] is an indispensable component in various natural language applications, such as story generation [7, 20], contextual text completion [18], and dialogue systems [22]. However, the conventional approach of training a language model with maximum likelihood estimation (MLE) and decoding the most likely sequence is often not sufficient [10, 27]. Specifically, this modelling formulation often leads to the problem of *degeneration*, i.e., the generated texts from the language model tend to be dull and contain undesirable repetitions at different levels (e.g., token-, phrase-, and sentence-level) [4]. To alleviate this problem, previous solutions modify the decoding strategy by sampling from *less* likely vocabularies [7, 10]. While reducing the generated repetition, these sampling methods introduce another critical problem (*semantic inconsistency*)—the sampled text tends to diverge from or even contradict to the original semantics defined by the human-written prefix [1]. Another approach addresses the degeneration problem by modifying the model's output vocabulary distribution with unlikelihood training [27].

In this work, we argue that the degeneration of neural language models stems from the *anisotropic* distribution of token representations, i.e., their representations reside in a narrow subset of the entire space [6, 5, 21]. In Figure 1(a), we showcase a cosine similarity matrix of token representations (taken from the output layer of the Transformer) produced by GPT-2. We see that the cosine similarities between tokens within a sentence are over **0.95**, meaning that these representations are close to each

---

[1]Our code and models are publicly available at https://github.com/yxuansu/SimCTG.

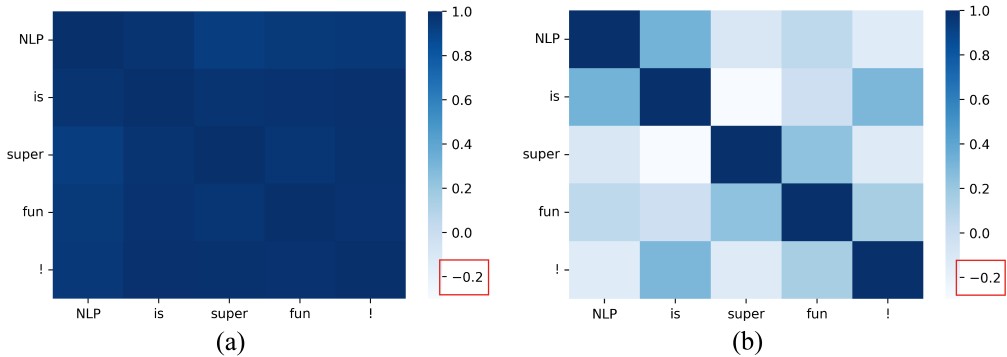

Figure 1: Token cosine similarity matrix of (a) GPT-2 and (b) SimCTG. (best viewed in color)

other. Such high similarity is undesirable as it can naturally cause the model to generate repetitive tokens at different steps. In an ideal setting, the token representations should follow an isotropic distribution, i.e., the token similarity matrix should be sparse and the representations of distinct tokens should be discriminative as shown in Figure 1(b). Moreover, during the decoding process, the sparseness of the token similarity matrix of the generated text should be preserved to avoid model degeneration.

Based on the above motivations, we present *SimCTG* (a **sim**ple **c**ontrastive framework for neural **t**ext **g**eneration) that encourages the model to learn discriminative and isotropic token representations. We also present a novel decoding strategy to complement SimCTG, *contrastive search*. The key intuitions behind contrastive search are: (i) at each decoding step, the output should be selected from the set of most probable candidates predicted by the model to better maintain the semantic coherence between the generated text and the human-written prefix, and (ii) the sparseness of the token similarity matrix of the generated text should be preserved to avoid degeneration.

We conduct comprehensive experiments on three widely used benchmarks. We show that our approach is generalizable to different tasks and different languages (§4 and §5) as well as different model sizes (§4.3 and Appendix D). Specifically, the experimental results verify that SimCTG improves the intrinsic qualities of the language model, as evaluated by perplexity and token prediction accuracy (§4.2 and Appendix D). Moreover, we demonstrate that the proposed contrastive search significantly outperforms previous state-of-the-art decoding methods in both human and automatic evaluations (§4 and §5). Furthermore, we provide in-depth analyses to get better insights on the inner-workings of our proposed approach (§6).

## 2  Background

### 2.1  Language Modelling

The goal of language modelling is to learn a probability distribution $p_\theta(\boldsymbol{x})$ over a variable-length text sequence $\boldsymbol{x} = \{x_1, ..., x_{|\boldsymbol{x}|}\}$, where $\theta$ denotes model parameters. Typically, the maximum likelihood estimation (MLE) objective is used to train the language model which is defined as

$$\mathcal{L}_{\text{MLE}} = -\frac{1}{|\boldsymbol{x}|} \sum_{i=1}^{|\boldsymbol{x}|} \log p_\theta(x_i|\boldsymbol{x}_{<i}). \tag{1}$$

However, as observed in many recent studies [6, 5, 21], training with likelihood maximization objective often yields an anisotropic distribution of model representations (especially for Transformer-based models) that undermines the model's capacity.

### 2.2  Open-ended Text Generation

In this work, we focus on studying the task of open-ended text generation due to its generality in various applications, such as story generation [7, 20], contextual text completion [18], poetry generation [14], and dialogue systems [22]. Formally, conditioned on a human-written prefix (i.e.,

context) $\boldsymbol{x}$, the task is to decode a continuation $\hat{\boldsymbol{x}}$ from the language model and the resulting text is $\{x_1, .., x_{|\boldsymbol{x}|}, \hat{x}_{|\boldsymbol{x}|+1}, ..., \hat{x}_{|\boldsymbol{x}|+|\hat{\boldsymbol{x}}|}\}$. Typically, there are two classes of methods used for decoding, which are (1) deterministic methods and (2) stochastic methods.

**Deteriminstic Methods.** Two widely used deterministic approaches are greedy and beam search which aim to select the text continuation with highest probability based on the model's probability distribution $p_\theta$. However, solely maximizing the output probability often leads to dullness [13] and degeneration [7, 10] in the generated text.

**Stochastic Methods.** To remedy the issues of deterministic decoding, several approaches have been proposed to sample from $p_\theta$. To avoid sampling from the unreliable tail of distribution, Fan *et al.* [7] proposed top-$k$ sampling which draws sample from the vocabulary subset $V^{(k)}$ that maximizes $\sum_{v \in V^{(k)}} p_\theta(v|\boldsymbol{x})$. Here, $|V^{(k)}| = k$ and $\boldsymbol{x}$ is the prefix context. Differently, the current state-of-the-art nucleus sampling [10] draws sample from the smallest vocabulary subset $U$ with total probability mass above a threshold $p \in [0, 1]$; i.e., $U$ is the smallest vocabulary subset such that $\sum_{v \in U} p_\theta(v|\boldsymbol{x}) \geq p$. While the sampling approaches help to alleviate model degeneration, the intrinsic stochasticity in these methods could cause the semantic meaning of the sampled text to diverge from or even contradict to the human-written prefix [1].

## 3 Methodology

In this section, we first present how to apply contrastive learning to calibrate the representation space of the language model. Then, we introduce our proposed contrastive search decoding algorithm.

### 3.1 Contrastive Training

Our goal is to encourage the language model to learn discriminative and isotropic token representations. To this end, we introduce a contrastive objective $\mathcal{L}_{\mathrm{CL}}$ into the training of the language model. Specifically, given a variable-length sequence $\boldsymbol{x} = \{x_1, ..., x_{|\boldsymbol{x}|}\}$, the $\mathcal{L}_{\mathrm{CL}}$ is defined as

$$\mathcal{L}_{\mathrm{CL}} = \frac{1}{|\boldsymbol{x}| \times (|\boldsymbol{x}| - 1)} \sum_{i=1}^{|\boldsymbol{x}|} \sum_{j=1, j \neq i}^{|\boldsymbol{x}|} \max\{0, \rho - s(h_{x_i}, h_{x_i}) + s(h_{x_i}, h_{x_j})\}, \tag{2}$$

where $\rho \in [-1, 1]$ is a pre-defined margin and $h_{x_i}$ is the representation of token $x_i$ produced by the model. The similarity function $s$ computes the cosine similarity between token representations as

$$s(h_{x_i}, h_{x_j}) = \frac{h_{x_i}^\top h_{x_j}}{\|h_{x_i}\| \cdot \|h_{x_j}\|}. \tag{3}$$

Intuitively, by training with $\mathcal{L}_{\mathrm{CL}}$, the model learns to pull away the distances between representations of distinct tokens.[2] Therefore, a discriminative and isotropic model representation space can be obtained. The overall training objective $\mathcal{L}_{\mathrm{SimCTG}}$ is then defined as

$$\mathcal{L}_{\mathrm{SimCTG}} = \mathcal{L}_{\mathrm{MLE}} + \mathcal{L}_{\mathrm{CL}}, \tag{4}$$

where the maximum likelihood estimation (MLE) objective $\mathcal{L}_{\mathrm{MLE}}$ is described in Eq. (1). Note that, when the margin $\rho$ in $\mathcal{L}_{\mathrm{CL}}$ equals to 0, the $\mathcal{L}_{\mathrm{SimCTG}}$ degenerates to the vanilla MLE objective $\mathcal{L}_{\mathrm{MLE}}$.

### 3.2 Contrastive Search

We propose a novel decoding method, *contrastive search*. At each decoding step, the key ideas of contrastive search are (i) the generated output should be selected from the set of most probable candidates predicted by the model; and (ii) the generated output should be discriminative enough with respect to the previous context. In this way, the generated text can (i) better maintain the semantic coherence with respect to the prefix while (ii) avoiding model degeneration.

Formally, given the previous context $\boldsymbol{x}_{<t}$, at time step $t$, the selection of the output $x_t$ follows

$$x_t = \underset{v \in V^{(k)}}{\arg\max} \left\{ (1 - \alpha) \times \underbrace{p_\theta(v|\boldsymbol{x}_{<t})}_{\text{model confidence}} - \alpha \times \underbrace{(\max\{s(h_v, h_{x_j}) : 1 \leq j \leq t - 1\})}_{\text{degeneration penalty}} \right\}, \tag{5}$$

---

[2]By definition, the cosine similarity $s(h_{x_i}, h_{x_i})$ of the identical token $x_i$ is 1.0.

where $V^{(k)}$ is the set of top-$k$ predictions from the model's probability distribution $p_\theta(\cdot|\boldsymbol{x}_{<t})$ and $k$ is typically set as 3~10. In Eq. (5), the first term, *model confidence*, is the probability of candidate $v$ predicted by the model. The second term, *degeneration penalty*, measures how discriminative of candidate $v$ with respect to the previous context $\boldsymbol{x}_{<t}$ and $s$ is defined in Eq. (3). Specifically, it is defined as the maximum cosine similarity between the representation of $v$ and that of all tokens in $\boldsymbol{x}_{<t}$. Here, the candidate representation $h_v$ is computed by the model given the concatenation of $\boldsymbol{x}_{<t}$ and $v$. Intuitively, a larger degeneration penalty of $v$ means it is more similar to the context, therefore more likely leading to model degeneration. The hyperparameter $\alpha \in [0, 1]$ regulates the importance of these two components. When $\alpha = 0$, contrastive search degenerates to the greedy search method.

## 4 Document Generation

We first evaluate our approach on the task of open-ended document generation.

**Model and Baselines.** Our proposed approach is architecture-agnostic and can be applied to any generation model. In this work, we evaluate our method on the representative GPT-2 model [18]. Specifically, we fine-tune GPT-2 on the evaluated benchmark (detailed below) with the proposed objective $\mathcal{L}_{\text{SimCTG}}$ (Eq. (4)) and generate the text continuation with different decoding methods. We perform experiments using the base model (117M parameters) which consists of 12 Transformer layers [25] with 12 attention heads.[3] We compare our approach with two strong baselines: (1) GPT-2 fine-tuned with the standard MLE objective (Eq. (1)); and (2) GPT-2 fine-tuned with unlikelihood objective [27].[4] Our implementation is based on the Huggingface Library [28].

**Evaluation Benchmark.** We conduct experiments on the Wikitext-103 dataset [16] which contains a large collection of Wikipedia articles with over 100 million words and 260 thousands unique tokens. Wikitext-103 is a document-level dataset and has been widely used for the evaluation of large-scale language modelling [3, 11, 29].

**Training.** For our SimCTG and the MLE baseline, we fine-tune the models on Wikitext-103 for 40k training steps. For the unlikelihood baseline, following Welleck *et al.* [27], we first fine-tune the model with the token-level unlikelihood objective for 38.5k steps and then with the sequence-level unlikelihood objective for 1.5k steps. Therefore, the overall training steps of all compared methods are the same. The batch size is set as 128 and the training samples are truncated to a maximum length of 256. We optimize the model with Adam optimizer [12] and a learning rate of 2e-5.

**Decoding.** We evaluate the models by producing text continuations given the prefixes from the test set. In the experiments, the lengths of the prefix and the generated continuation are set as 32 and 128, respectively. We test different models with various decoding methods. For deterministic method, we use greedy search and beam search with a beam size of 10. For stochastic method, we use the current state-of-the-art nucleus sampling [10] with $p = 0.95$. For the proposed contrastive search, the $k$ and $\alpha$ in Eq. (5) are set as 8 and 0.6.[5] The hyperparameters of different methods are selected based on their optimal MAUVE (detailed in §4.1.2) performance on the validation set.

### 4.1 Evaluation Metrics

We perform evaluation from two aspects: (1) *language modelling quality* which measures the intrinsic quality of the model; and (2) *generation quality* which measures the quality of the generated text.

#### 4.1.1 Language Modelling Quality

Following Welleck *et al.* [27], we report the results of the model on the metrics below.

**Perplexity.** The model perplexity (**ppl**) on the test set of Wikitext-103.

**Prediction Accuracy.** It is defined as: $\textbf{acc} = \frac{1}{\sum_{\boldsymbol{x}\in\mathcal{D}}|\boldsymbol{x}|}\sum_{\boldsymbol{x}\in\mathcal{D}}\sum_{t=1}^{|\boldsymbol{x}|}\mathbb{1}[\arg\max p_\theta(x|\boldsymbol{x}_{<t}) = x_t]$, where $\mathcal{D}$ is the Wikitext-103 test set, $\boldsymbol{x}_{<t}$ is the prefix, and $x_t$ is the reference token at time step $t$.

---

[3]In Appendix D, we demonstrate the experimental results of our approach on other language models.
[4]The unlikelihood baseline is implemented with the official code, which can be found at `https://github.com/facebookresearch/unlikelihood_training`.
[5]In Appendix E, we provide detailed ablation studies on the effect of both $k$ and $\alpha$ in contrastive search.

| Model | Language Modelling Quality | | | | Generation Quality | | | | | | | |
|---|---|---|---|---|---|---|---|---|---|---|---|---|
| | ppl↓ | acc↑ | rep↓ | wrep↓ | Method | rep-2↓ | rep-3↓ | rep-4↓ | diversity↑ | MAUVE↑ | coherence↑ | gen-ppl |
| MLE | 24.32 | 39.63 | 52.82 | 29.97 | greedy | 69.21 | 65.18 | 62.05 | 0.04 | 0.03 | 0.587 | 7.32 |
| | | | | | beam | 71.94 | 68.97 | 66.62 | 0.03 | 0.03 | 0.585 | 6.42 |
| | | | | | nucleus | 4.45 | 0.81 | 0.43 | 0.94 | 0.90 | 0.577 | 49.71 |
| | | | | | contrastive | 44.20 | 37.07 | 32.44 | 0.24 | 0.18 | 0.599 | 9.90 |
| Unlike. | 28.57 | 38.41 | **51.23** | **28.57** | greedy | 24.12 | 13.35 | 8.04 | 0.61 | 0.69 | 0.568 | 37.82 |
| | | | | | beam | 11.83 | 5.11 | 2.86 | 0.81 | 0.75 | 0.524 | 34.73 |
| | | | | | nucleus | 4.01 | 0.80 | 0.42 | **0.95** | 0.87 | 0.563 | 72.03 |
| | | | | | contrastive | 7.48 | 3.23 | 1.40 | 0.88 | 0.83 | 0.574 | 43.61 |
| SimCTG | **23.82** | **40.91** | 51.66 | 28.65 | greedy | 67.36 | 63.33 | 60.17 | 0.05 | 0.05 | 0.596 | 7.16 |
| | | | | | beam | 70.32 | 67.17 | 64.64 | 0.04 | 0.06 | 0.591 | 6.36 |
| | | | | | nucleus | 4.05 | 0.79 | 0.37 | 0.94 | 0.92 | 0.584 | 47.19 |
| | | | | | contrastive | **3.93** | **0.78** | **0.31** | **0.95** | **0.94** | **0.610** | **18.26** |
| Human | - | - | 36.19 | - | - | 3.92 | 0.88 | 0.28 | 0.95 | 1.00 | 0.644 | 24.01 |

Table 1: Evaluation results on Wikitext-103 test set. "Unlike." denotes the model trained with unlikelihood objective. ↑ means higher is better and ↓ means lower is better.

**Prediction Repetition.** The fraction of next-token (top-1) predictions that occur in the prefix which is defined as: $\mathbf{rep} = \frac{1}{\sum_{\boldsymbol{x} \in \mathcal{D}} |\boldsymbol{x}|} \sum_{\boldsymbol{x} \in \mathcal{D}} \sum_{t=1}^{|\boldsymbol{x}|} \mathbb{1}[\arg\max p_\theta(x|\boldsymbol{x}_{<t}) \in \boldsymbol{x}_{<t}]$.

In addition, the next token repetitions that do not equal to the ground truth token: $\mathbf{wrep} = \frac{1}{\sum_{\boldsymbol{x} \in \mathcal{D}} |\boldsymbol{x}|} \sum_{\boldsymbol{x} \in \mathcal{D}} \sum_{t=1}^{|\boldsymbol{x}|} \mathbb{1}[\arg\max p_\theta(x|\boldsymbol{x}_{<t}) \in \boldsymbol{x}_{<t} \wedge \neq x_t]$ is also reported.

### 4.1.2 Generation Quality

**Generation Repetition.** This metric measures the sequence-level repetition as the portion of duplicate $n$-grams in the generated text [27]. For a generated text continuation $\hat{\boldsymbol{x}}$, the repetion at $n$-gram level is defined as: $\mathbf{rep\text{-}n} = 100 \times (1.0 - \frac{|\text{unique n-grams}(\hat{\boldsymbol{x}})|}{|\text{total n-grams}(\hat{\boldsymbol{x}})|})$.

**Diversity.** This metric takes into account the generation repetition at different $n$-gram levels and it is defined as: $\mathbf{diversity} = \prod_{n=2}^{4}(1.0 - \frac{\text{rep-n}}{100})$. It can be deemed as an overall assessment of model degeneration. A lower diversity means a more severe degeneration of the model.

**MAUVE** [17] is a metric that measures the token distribution closeness between the generated text and human-written text. A higher MAUVE score means the model generates more human-like texts.

**Semantic Coherence.** To automatically measure the semantic coherence (i.e., consistency) between the prefix and the generated text, we employ the advanced sentence embedding method, SimCSE [9]. Specifically, given the prefix $\boldsymbol{x}$ and the generated text $\hat{\boldsymbol{x}}$, the coherence score is defined as: $\mathbf{coherence} = v_{\boldsymbol{x}}^\top v_{\hat{\boldsymbol{x}}}/(\|v_{\boldsymbol{x}}\| \cdot \|v_{\hat{\boldsymbol{x}}}\|)$, where $v_{\boldsymbol{x}} = \text{SimCSE}(\boldsymbol{x})$ and $v_{\hat{\boldsymbol{x}}} = \text{SimCSE}(\hat{\boldsymbol{x}})$.

**Perplexity of Generated Text.** Lastly, we evaluate the perplexity of the generated text $\hat{\boldsymbol{x}}$ given the prefix $\boldsymbol{x}$, which is defined as: $\mathbf{gen\text{-}ppl} = 2^{f(\mathcal{D},\theta)}$ and $f(\mathcal{D},\theta) = \frac{1}{\sum_{\boldsymbol{x} \in \mathcal{D}} |\hat{\boldsymbol{x}}|} \sum_{\boldsymbol{x} \in \mathcal{D}} \log_2 p_\theta(\hat{\boldsymbol{x}}|\boldsymbol{x})$.

Importantly, the optimal approach should produce text which has a perplexity *close* to that of the human-written text [10]. A high gen-ppl means the generated text is very *unlikely* given the prefix, therefore being low quality. In contrastive, a low gen-ppl means the generated text has a low diversity and gets stuck in repetitive loops [10]. We use the model $\theta$ trained with $\mathcal{L}_{\text{SimCTG}}$ to measure the gen-ppl of different approaches, therefore making sure the numbers are comparable with each other.[6]

### 4.2 Results

The experimental results on Wikitext-103 are shown in Table 1.

**Language Modelling Quality.** From the results, we observe that SimCTG achieves the best perplexity and next token accuracy. The reason is that, with more discriminative representations, SimCTG is less confusing when making next token predictions, leading to the improved model performance.

---

[6]We obtain similar gen-ppl results and can draw the same conclusion when using the model trained with MLE and Unlikelihood. Therefore, we only include the results acquired by the model trained with $\mathcal{L}_{\text{SimCTG}}$ in Table 1. We refer to Appendix F for the gen-ppl results obtained by the MLE and Unlikelihood models.

On the rep and wrep metrics, the unlikelihood model yields the best result but at the expense of unfavorable performance drops in the perplexity and next token accuracy.

**Generation Quality.** Firstly, on the rep-n and diversity metrics, SimCTG + contrastive search obtains the best result, suggesting it best addresses the degeneration problem. Secondly, the MAUVE score demonstrates that SimCTG + contrastive search generates texts that are closest to human-written texts in terms of token distribution. Thirdly, among all methods, SimCTG + contrastive search is the only approach that achieves over 0.6 coherence score, showing it produces semantically consistent text with respect to the prefix. Lastly, the gen-ppl metric also validates the superiority of SimCTG + contrastive search as it obtains notably better generation perplexity comparing with other approaches.

Moreover, from the results of MLE and Unlikelihood baselines, we see that contrastive search still brings performance boost as compared with greedy and beam search. However, the performance gain still lags behind SimCTG, which demonstrates the necessity of contrastive training. The underlying reason is that, without using the contrastive objective $\mathcal{L}_{\text{CL}}$ (Eq. (2)), the token representations obtained by MLE or Unlikelihood are less discriminative (§6.1). Therefore, the degeneration penalty (Eq. (5)) of different candidates are less distinguishable and the selection of output is dominated by the model confidence, making contrastive search less effective.

| Model | Decoding Method | Coherence | Fluency | Informativeness |
|---|---|---|---|---|
| Agreement | - | 0.51 | 0.64 | 0.70 |
| MLE | nucleus | 2.92 | 3.32 | 3.91 |
| | contrastive | 2.78 | 2.29 | 2.56 |
| Unlikelihood | nucleus | 2.59 | 3.02 | 3.58 |
| | contrastive | 2.76 | 2.90 | 3.35 |
| SimCTG | nucleus | 2.96 | 3.34 | 3.96 |
| | contrastive | 3.25★ | 3.57★ | 3.96 |
| SimCTG-large | nucleus | 3.01 | 3.37 | **3.98** |
| | contrastive | **3.33**★ | **3.66**★ | **3.98** |
| Human | - | 3.70 | 3.71 | 4.21 |

Table 2: Human evaluation results. ★ results significantly outperforms the results of nucleus sampling with different models (Sign Test with p-value < 0.05).

## 4.3 Human Evaluation

We also conduct a human evaluation with the help of graders proficient in English from a third-party grading platform. We randomly select 200 prefixes with length of 32 from the test set of Wikitext-103. For each prefix, we use different models (MLE, Unlikelihood, and SimCTG) with two decoding methods (nucleus sampling and contrastive search) to generate text continuations with length of 128. To examine the generality of our approach across different model sizes, we include a large size SimCTG (i.e., **SimCTG-large**) which is obtained by fine-tuning the GPT-2-large model that consists of 36 Transformer layers with 20 attention heads. All generated results, plus the reference text, are randomly shuffled and evaluated by five graders, which results in 9,000 annotated samples in total. The evaluation follows a 5-point Likert scale (1, 2, 3, 4, or 5) for each of the following features:[7]

- **Coherence**: Whether the generated text is semantically consistent with the prefix.
- **Fluency**: Whether the generated text is fluent and easy to understand.
- **Informativeness**: Whether the generated text is diverse and contains interesting content.

Table 2 presents the human evaluation results, with the first row showing strong inter-annotator agreements as measured by Fleiss′ kappa coefficient [8]. Firstly, we see that, directly applying contrastive search with MLE or Unlikelihood model does not yield satisfactory results. This is due to the anisotropic nature of their representation space as discussed in Section §4.2. Secondly, the coherence score of Unlikelihood model is notably lower than MLE and SimCTG, suggesting it generates the most *unlikely* results which is also shown by its generation perplexity (gen-ppl) in Table 1. Furthermore, the results of SimCTG + contrastive search significantly outperforms nucleus sampling with different models in terms of coherence and fluency (Sign Test with p-value < 0.05).

---

[7]We refer to Appendix G for more details of human evaluation.

Lastly, SimCTG-large + contrastive search achieves the best performance across the board and even performs comparably with human-written text on the fluency metric (Sign Test with p-value > 0.4). This reveals the clear generalization ability of our approach to large size models and future work could focus on extending it to models that contain over billions of parameters such as GPT-3 [2].

## 5   Open-domain Dialogue Generation

To test the generality of our approach across different tasks and languages, we then evaluate our method on the task of open-domain dialogue generation. In this task, given a multi-turn dialogue context (where each turn is an user utterance), the model is asked to generate an adequate response that is semantically consistent with the context. Here, the dialogue context is deemed as the prefix.

**Benchmark and Baselines.** We conduct experiments on two benchmark datasets from two languages (i.e., Chinese and English). For the Chinese benchmark, we use the LCCC dataset [26]. For the English Benchmark, we use the DailyDialog dataset [15].

We compare the GPT-2 models fine-tuned with SimCTG and MLE.[8] Specifically, for the Chinese benchmark (i.e., LCCC), we use a publicly available Chinese GPT-2 [31].[9] Same as in Section §4, during training, we use a batch size of 128 and truncate the training samples to a maximum length of 256. On the LCCC dataset, we train (i.e., fine-tune) the models for 40k steps. As for the DailyDialog dataset, due to its smaller dataset size, we train the models for 5k steps. For optimization, we use Adam optimizer and a learning rate of 2e-5.

For each model, we use four decoding methods, including (1) greedy search; (2) beam search (beam size of 10); (3) nucleus sampling ($p = 0.95$); and (4) contrastive search ($k = 5$, $\alpha = 0.6$).

**Evaluation.** We rely on human evaluation to assess the model performance. Same as in Section §4.3, we randomly select 200 dialogue contexts from the test set and ask five annotators to evaluate the generated responses plus the reference response in three dimensions: (i) coherence, (ii) fluency; and (iii) informativeness. The scores follow a 5-point Likert scale (1, 2, 3, 4, or 5).

| Model | Method | LCCC | | | DailyDialog | | |
|-------|--------|------|------|------|------|------|------|
| | | Coherence | Fluency | Informativeness | Coherence | Fluency | Informativeness |
| Agreement | - | 0.73 | 0.61 | 0.57 | 0.64 | 0.60 | 0.55 |
| MLE | greedy | 3.01 | 3.27 | 1.97 | 3.28 | 3.51 | 2.92 |
| | beam | 2.60 | 2.90 | 1.55 | 3.16 | 3.43 | 2.78 |
| | nucleus | 2.78 | 3.55 | 2.64 | 2.67 | 3.58 | 3.42 |
| | contrastive | 3.28★ | 3.84★ | 3.06★ | 3.27 | 3.41 | 2.82 |
| SimCTG | greedy | 3.04 | 3.32 | 2.01 | 3.31 | 3.50 | 2.94 |
| | beam | 2.57 | 2.93 | 1.59 | 3.19 | 3.45 | 2.79 |
| | nucleus | 2.84 | 3.58 | 2.72 | 2.75 | 3.59 | 3.39 |
| | contrastive | **3.32★** | **3.96★** | **3.13★** | **3.73★** | **3.85★** | **3.46** |
| Human | - | 3.42 | 3.76 | 3.20 | 4.11 | 3.98 | 3.74 |

Table 3: Human evaluation results. ★ results significantly outperforms the results of greedy search, beam search, and nucleus sampling with different models. (Sign Test with p-value < 0.05).

Table 3 shows the evaluation results where the first row shows strong inter-annotator agreements as measured by Fleiss′ kappa coefficient. On both datasets, we see that SimCTG + contrastive search significantly outperforms other methods on various metrics, suggesting that our approach is generalizable to different languages and tasks. It is worth emphasizing that, on the LCCC benchmark, SimCTG + contrastive search surprisingly outperforms the human performance on the fluency metric, while performing comparably on the coherence and informativeness metrics (Sign Test with p-value > 0.4). Moreover, even **without** contrastive training, the MLE model performs significantly better when using contrastive search. This is due to the intrinsic property of Chinese language model for which the MLE objective can already yield a representation space that displays a high level of isotropy,

---

[8]We acknowledge that there are other GPT-like models (e.g., Zhang *et al.* [30] and Thoppilan *et al.* [24]) that are designed for dialogue generation. We leave the test of our approach on these models to our future work.

[9]https://huggingface.co/uer/gpt2-chinese-cluecorpussmall

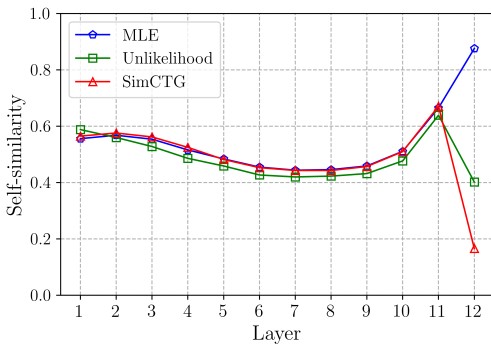
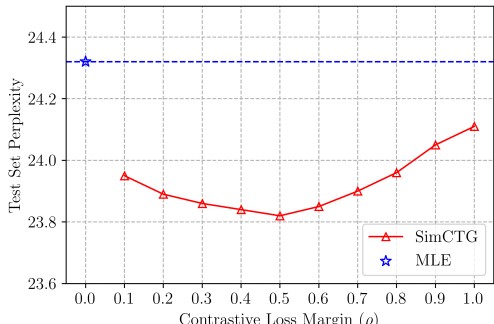

Figure 2: Layer-wise representation self-similarity.    Figure 3: The effect of contrastive margin $\rho$.

making contrastive search directly applicable.[10] This finding is particularly attractive as it reveals the potential applicability of contrastive search on off-the-shelf (i.e., without contrastive training) language models for certain languages such as Chinese.

## 6 Further Analysis

### 6.1 Token Representation Self-similarity

To analyze the token representations learned by SimCTG, we follow Ethayarajh [6] and define the averaged self-similarity of token representations within a text sequence $\boldsymbol{x}$ as

$$\text{self-similarity}(\boldsymbol{x}) = \frac{1}{|\boldsymbol{x}| \times (|\boldsymbol{x}|-1)} \sum_{i=1}^{|\boldsymbol{x}|} \sum_{j=1, j\neq i}^{|\boldsymbol{x}|} \frac{h_{x_i}^{\top} h_{x_j}}{\|h_{x_i}\| \cdot \|h_{x_j}\|}, \tag{6}$$

where $h_{x_i}$ and $h_{x_j}$ are the token representations of $x_i$ and $x_j$ produced by the model. Intuitively, a lower self-similarity($\boldsymbol{x}$) indicates the representations of distinct tokens within the sequence $\boldsymbol{x}$ are less similar to each other, therefore being more discriminative.

We use texts from Wikitext-103 test set and compute the self-similarity of token representations over different layers for different models. Figure 2 plots the results averaged over all samples. We see that, in the intermediate layers, the self-similarity of different models are relatively the same. In contrast, at the output layer (layer 12), SimCTG's self-similarity becomes notably lower than other baselines. We note that the Unlikelihood model also yields more discriminative representations than MLE, but its language model accuracy is lower than MLE and SimCTG as shown in Table 1. On the other hand, SimCTG obtains the most discriminative and isotropic representations while maintaining the best language model accuracy, which further validates the clear advantage of our proposed approach.

### 6.2 The Effect of Contrastive Loss Margin

Next, we analyze the effect of contrastive loss margin $\rho$ (Eq. (2)). To this end, we fine-tune the GPT-2 by varying $\rho$ from 0.1 to 1.0 and measure the model perplexity on the Wikitext-103 test set. Figure 3 plots the results of different $\rho$ along with the result of the MLE baseline. Note that, when $\rho = 0$, SimCTG is equivalent to MLE (Section §3.1). From Figure 3, we see that the contrastive training always helps to improve the perplexity as compared with MLE. However, when $\rho$ is either too small (e.g., 0.1) or large (e.g., 1.0), the learned representation space of the model would be either less or too isotropic, leading to a sub-optimal perplexity. In our experiments, the most suitable margin $\rho = 0.5$.

### 6.3 Contrastive Search versus Nucleus Sampling

Then, we provide an in-depth comparsion between our proposed contrastive search and the current state of the art, nucleus sampling. To this end, we compare the results of SimCTG using these two decoding methods. Specifically, we vary the probability $p$ for nucleus sampling and the $\alpha$ (Eq. (5))

---

[10]We provide more in-depth analyses and several generated examples on LCCC in Appendix H and J, respectively.

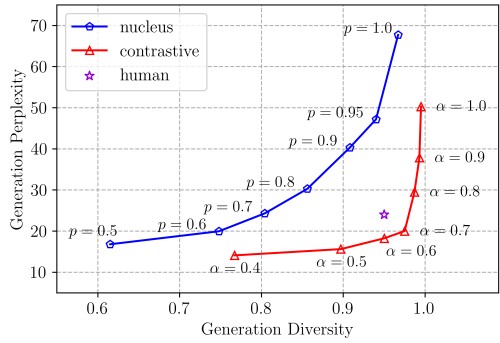
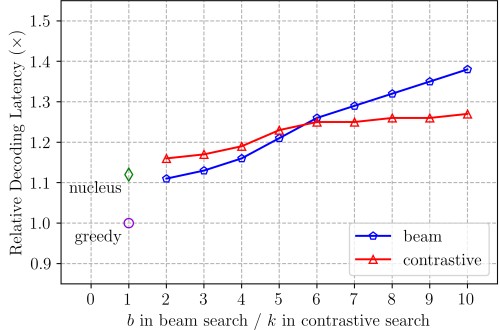

Figure 4: Contrastive search vs nucleus sampling.   Figure 5: Inference latency comparison.

for contrastive search to generate results using prefixes from Wikitext-103 test set.[11] We evaluate the results from two aspects: (1) generation diversity and (2) perplexity of the generated text (gen-ppl). Both metrics are described in Section §4.1.2. Figure 4 plots the results of different methods along with the human performance. For nucleus sampling, when $p$ is small (i.e., $p \leq 0.7$), its generation perplexity is comparable to that of human. However, the diversity is notably lower than human performance, meaning it stuck in undesirable repetition loops [10]. On the other hand, when $p$ is large (i.e., $p \geq 0.95$), the generation diversity is close to that of human but the generation perplexity is significantly higher. Such high perplexity means the generated text is very *unlikely*, therefore being low quality. As for contrastive search, when $\alpha \in [0.5, 0.8]$, it yields generation diversity and perplexity that are both comparable to human performance. These results demonstrate the superiority of contrastive search as it better balances the trade-off between the generation diversity and perplexity.

## 6.4 Decoding Latency Comparison

We compare the decoding latency of different decoding methods using SimCTG. For beam search and contrastive search, we vary the beam width $b$ and the $k$ in Eq. (5). The latency is measured by generating fixed length text continuations on Wikitext-103 test cases with a batch size of 1. In Figure 5, we show the averaged relative decoding latency of different methods. We see that greedy search is the fastest method and the latency of different methods are generally comparable with each other. Comparing contrastive search with beam search, when $b$ and $k$ are small (i.e., $\leq 6$), their latency are nearly identical. When $b$ and $k$ gets larger (i.e., $> 6$), contrastive search becomes faster. In summary, these comparison results further verify the practical usage of contrastive search.

| | prefix | div. | coh. | gen-ppl |
|---|---|---|---|---|
| **prefix** | Butt criticized Donald's controls in certain situations in the game, as well as the difficulty of some levels and puzzles. Buchanan also criticized the controls, calling | | | |
| beam | them " too easy " and " too difficult ". He also criticized the game's graphics, stating that the game's graphics were " too simplistic " and " too simplistic ". He also criticized the game's graphics, stating that the game's graphics were " too simplistic " and " too simplistic ". He also criticized the graphics, stating that the game's graphics were " too simplistic " and " too simplistic ". He also criticized the graphics, stating that the game's graphics were " too simplistic " and " too simplistic ". He also criticized the graphics, stating that the game's graphics were" ... | 0.02 | 0.605 | 7.91 |
| nucleus | them " offensively rudimentary ". After the game was released, Buchanan wrote a number of game reviews about his review of the game, primarily summarizing what he thought the reviewers had enjoyed about it. Buchanan was pleased with how consistently these reviews awarded the game. For one, he praised the puzzles, compared them to Nos. 1 and 2 and enjoyed the construction of the artificial intelligence in the game's forms. He found no fault with the puzzles, even in those situations where they seemed repetitive. According to an Xbox Magazine review, " every time ... | 0.98 | 0.597 | 37.83 |
| contrastive | them " unimpressive " and a " nightmare " of an experience to play with players unfamiliar with Tetris. On the other hand, his opinion was shared by other reviewers, and some were critical of the game's technical design for the Wii version of Tetris. In addition, Tintin's review included a quote from Roger Ebert, who said that Tetris was better than the original game due to its simplicity and ease of play. Ebert's comments were included in the game's DVD commentary, released on March 22, 2010. It is unclear if any of the video commentary was taken from ... | 0.98 | 0.626 | 19.64 |

Table 4: **Case Study**: The beam search produces degeneration repetitions (highlighted in red) and the nucleus sampling produces text that has incoherent semantics with respect to the prefix (highlighted in blue). The reasonable repetitions produced by contrastive search are highlighted in green. The "div." and "coh." stand for diversity and coherence metrics. (best viewed in color)

---

[11]For contrastive search, we only vary the value of $\alpha$ and keep $k$ constant to 8 as described in Section §4. In Appendix E, we provide detailed ablation studies on the effect of both $k$ and $\alpha$ in contrastive search.

## 6.5 Case Study

In Table 4, we present generated examples of SimCTG with different decoding methods given a specific prefix.[12] From the results, we see that beam search produces undesirable sequence-level repetitions, resulting in low diversity and low generation perplexity. On the other hand, in the prefix, the person "Buchanan" *criticizes* the game. However, the result from nucleus sampling displays a contradicted semantic, resulting in a low coherence score as well as a high generation perplexity. As for contrastive search, it generates a text that is semantically consistent to the prefix with a proper generation perplexity while obtaining the same diversity as that of the nucleus sampling. Additionally, it is worth emphasizing that, while the degeneration penalty in Eq. (5) encourages the model to generate diverse outputs, contrastive search is still able to generate reasonable repetitions as highlighted in Table 4. This is due to the incorporation of model confidence in Eq. (5) which enables the model to repeat the important content (e.g., person names or entity names) from the previous context like humans do.

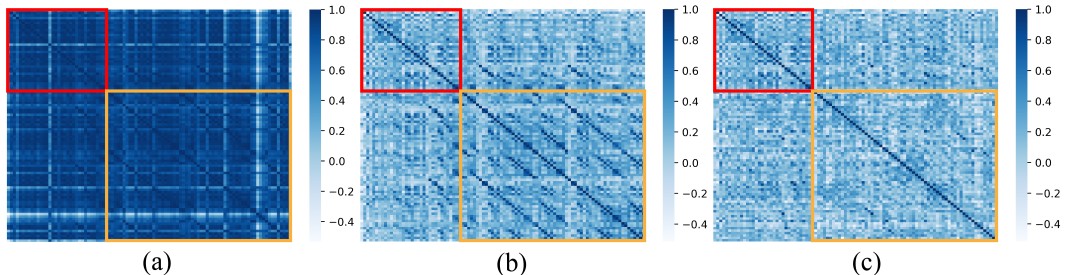

Figure 6: (a) MLE + beam search; (b) SimCTG + beam search; (c) SimCTG + contrastive search. The token similarity matrix of the prefix and the generated text are highlighted in red and yellow.

## 6.6 Comparison of Token Similarity Matrix

To better understand how contrastive search works, in Figure 6, we show the generated token similarity matrix of SimCTG using beam search and contrastive search. For a better comparsion, we also include the result of MLE using beam search. All results are produced with the same prefix as in Table 4. The red and yellow boxes highlight the similarity matrix of the prefix and the generated text. Firstly, we see that, the MLE + beam search yields a very dense similarity matrix, meaning that its token representations are indiscriminative. In addition, the high similarity scores in its off-diagonal entries clearly show the degeneration repetitions. Secondly, for SimCTG + beam search, we observe a desirable similarity matrix of the prefix which is sparse and isotropic. However, degeneration repetitions still exist in the generated result as shown in Figure 6(b). Lastly, for SimCTG + contrastive search, the entire similarity matrix is sparse and isotropic, showing that it successfully solves the model degeneration. These observations are in line with our motivations as described in Section §1.

## 7 Conclusion

In this work, we show that the degeneration of neural language models stems from the anisotropic nature of their token representations. We present a new approach, *SimCTG*, for training the language model such that it obtains an isotropic and discriminative representation space. In addition, we introduce a novel decoding method, *contrastive search*, which works coherently with the proposed SimCTG. Extensive experiments and analyses are conducted on three benchmarks from two languages. Both automatic and human evaluations demonstrate that our approach substantially reduces model degeneration and significantly outperforms current state-of-the-art text generation approaches.

## Acknowledgments

The first author would like to thank Jialu Xu and Huayang Li for their insightful discussions and supports. Many thanks to our anonymous reviewers, area chairs, and senior area chairs for their suggestions and comments.

---

[12]We refer to Appendix K for more generated examples of SimCTG.

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
