# OpenReview forum: "A Contrastive Framework for Neural Text Generation"
_NeurIPS.cc/2022/Conference — NeurIPS 2022 Accept_

### Official Review · Reviewer_v9y3 · 2022-07-03

**Rating:** 7
**Confidence:** 4
**Soundness:** 4 excellent
**Presentation:** 4 excellent
**Contribution:** 2 fair

**Summary:**

Current language models suffer from the issue of degeneration, which makes their outputs dull and repetitive. This paper argues that this degeneration in text generation is due to the anisotropic distribution of LMs. Prior work [1] has shown that LM representations are anisotropic --- the cosine similarity between the representation vectors for *different* tokens in a sentence is very high, up to 0.95. This paper designs training + inference time algorithms to fix this issue, and report improvements in text generation quality.

This paper proposes the SimCTG algorithm, which helps make representations more isotropic during training. The key idea is to use an extra loss function which pushes away representations of different tokens. During inference, the paper proposes the use of contrastive search, a decoding objective which encourages the generation of tokens whose representations are dissimilar to one another. Contrastive search can be applied without the contrastive training on any existing LM.

The paper evaluates their approach on open-ended dialogue and document generation, covering both English and Chinese datasets. Extensive automatic and human confirm that the proposed approach beats baselines like nucleus sampling & unlikelihood training.

[1] - https://aclanthology.org/D19-1006.pdf

**Questions:**

None

**Limitations:**

Good limitations section in Appendix

**Strengths And Weaknesses:**

**Strengths**

1. The paper works towards fixing an important issue in current text generation systems --- their tendency to produce degenerate text. The paper draws an nice connection to prior work [1] which has found LM representations are anisotropic.

2. The paper presents simple training and inference time algorithms to make language model representations more isotropic. The inference time algorithm (contrastive search) can be flexibly applied to any existing LM without further training, and seems to improve over nucleus sampling for dialogue generation even without contrastive training.

3. The paper evaluates their algorithm on three datasets spanning two tasks and two languages. Extensive automatic evaluation (using effective metrics like MAUVE) and human evaluation is conducted to confirm the efficacy method.

4. The main experiments are complemented with good analysis experiments discussing qualitative aspects of improvements with model generations, timing analysis, effect of varying hyperparameters, isotropy of SimCTG representations.

**Weaknesses**

I have one major concern with the paper. All experiments have been conducted on a very small language model, GPT2-small which has 117M parameters. However, GPT2-small is a weak language model compared to much larger open-source alternatives like GPT2-large / XL [6], T5 variants fine-tuned for causal language modeling [2], OPT [5] etc. For dialogue generation you could simply fine-tune larger T5 variants or BART.

Testing out novel text generation ideas at a larger scale is important --- as the LMs get bigger, generative quality of MLE baselines significantly improves. It's unclear how anisotropic larger LMs are, and if yes, whether this anisotropy affects generation quality. A related informal tweet - [7].

**Minor**

It will be good to discuss the relations to [3, 4] in the related work since they use contrastive learning for text generation as well. These two papers have come out in the last month (contemporary to this paper), so no direct comparison is necessary.

In the introduction, explicitly mentioned that the anisotropic representation issue was first noticed by [1].

In Table 2 / 3, keep a column which aggregates scores and mentions the fraction of data points where all three metrics simultaneously score a 4 or 5. This will give a good idea about the overall performance of methods.

line 173: confusing --> confused

**Overall** --- This paper has good originality, clarity, quality, but moderate significance. My main concern is that experiments were only conducted on GPT2-small, a very small language model compared to the state-of-the-art. Nevertheless, I'm leaning accept due to interesting idea as well as good automatic & human evaluation conducted in the paper.

---

**After Rebuttal** - Thank you for your comprehensive reply and addressing the weakness I pointed out. I've raised my score to 7.

[1] - https://aclanthology.org/D19-1006.pdf
[2] - https://github.com/google-research/text-to-text-transfer-transformer/blob/main/released_checkpoints.md#lm-adapted-t511lm100k
[3] - https://arxiv.org/abs/2205.09726
[4] - https://arxiv.org/abs/2205.14690
[5] - https://arxiv.org/pdf/2205.01068.pdf
[6] - https://d4mucfpksywv.cloudfront.net/better-language-models/language_models_are_unsupervised_multitask_learners.pdf
[7] - https://mobile.twitter.com/_jasonwei/status/1526589104758042624

---

> ### Author Response · Authors · 2022-08-01
> **Response to the Reviewer v9y3:**
>
> #### 1.2. How larger language models perform on open-ended text generation with the proposed approach:
> |Model|Model Size| Objective|ppl$\\downarrow$|acc$\\uparrow$|conicity$\\downarrow$|self-similarity$\\downarrow$|Method|diversity$\\uparrow$|MAUVE$\\uparrow$|coherence$\\uparrow$|
> |:-------------:|:-------------:|:-------------:|:-----------------------:|:-----------------------:|:-----------------------:|:-----------------------:|:-----------------------:|:-----------------------:|:-----------------------:|:-----------------------:|
> |**Vanilla Transformer**|117M|MLE|26.60|35.62|0.50|0.22|nucleus|0.89|0.81|0.541|
> ||||||||contrastive|0.90|0.83|0.561|
> |||SimCTG|**26.55**|**36.03**|**0.47**|**0.19**|nucleus|0.89|0.82|0.543|
> ||||||||contrastive|**0.91**|**0.85**|**0.566**|
> ||||||||||||
> |**GPT-2-small**|117M|MLE|24.32|39.63|0.90|0.86|nucleus|0.94|0.90|0.577|
> ||||||||contrastive|0.24|0.18|0.599|
> |||SimCTG|**23.82**|**40.91**|**0.43**|**0.18**|nucleus|0.94|0.92|0.584|
> ||||||||contrastive|**0.95**|**0.94**|**0.610**|
> ||||||||||||
> |**GPT-2-large**|774M|MLE|16.57|43.34|0.46|0.20|nucleus|0.94|0.91|0.583|
> ||||||||contrastive|**0.95**|**0.96**|0.623|
> |||SimCTG|**16.53**|**43.47**|**0.42**|**0.17**|nucleus|**0.95**|0.93|0.591|
> ||||||||contrastive|**0.95**|**0.96**|**0.626**|
> ||||||||||||
> |**Human**|-|-|-|-|-|-|-|0.95|1.00|0.644|
>
> Next, we conduct experiments on Wikitext-103 and evaluate the performance of larger language models on open-ended text generation. Specifically, we use three different underlying language models, including vanilla transformer (177M), GPT-2-small (177M), and GPT-2-large (774M). The experimental results are shown in the Table above where the results of GPT-2-small are copied from Table 1 of our paper.
>
> From the results, we have the following findings:
>   * (1) SimCTG + contrastive search always outperforms the strongest baseline (MLE + nucleus sampling) under all model configurations.
>   * (2) When the language model gets large enough, the contrastive search can achieve superior performance even **without** SimCTG. This is due to the fact that larger language models can better learn isotropic representation space as we demonstrated in our answer \#1.1.
>   * (3) The performance of GPT-2-large with SimCTG + contrastive search is also in line with our human evaluation results provided in the main paper. In Table 2 of our paper, we show that GPT-2-large performs better than GPT-2-small when using SimCTG + contrastive search (Line 211). These results validate the clear generalization ability of our approach to larger language models.
>
> To summarize,
>   * (i) When the size of the language model gets large enough (e.g. 774M parameters for GPT-2-large), the contrastive search can be directly applied to **off-the-shelf** language models and can yield the best generation performances **without** any additional training.
>   * (ii) Under the circumstances where the computational overhead and inference latency are the primary concerns, smaller language models are always preferred. In such cases, SimCTG is a simple yet effective solution to boost the performance of smaller language models.
>
> In the camera-ready version, we will add some discussions on the performance of our approach on larger language models. We will save the full-scope and rigorous investigations for our future work. Thank you again for suggesting this interesting and definitely valuable research direction!
>
> ### 2. Suggested modifications:
>   * (1) Related references: Thank you for sharing these interesting and related references with us. We will add them in our camera-ready version.
>   * (2) Introduction: We will adjust our writing in the introduction and explicitly mention that the anisotropic representation issue was first noticed by [2].
>   * (3) Aggregated human evaluation score: Thank you for your suggestion. In the camera-ready version, we will add another column of aggregated human evaluation score to better present our results.
>   * (4) Typo: We will fix the typo in our next version. Thank you for pointing it out!
>
>
> [1] - [https://aclanthology.org/P18-1012/](https://aclanthology.org/P18-1012/)
>
> [2] - [https://aclanthology.org/D19-1006.pdf](https://aclanthology.org/D19-1006.pdf)

---

> > ### Comment · Reviewer_v9y3 · 2022-08-07
> > **Thank you, score raised to 7**
> >
> > Thank you for your comprehensive reply and addressing the weakness I pointed out with experiments on larger variants of GPT2. I've raised my score to 7.

---

> > > ### Author Response · Authors · 2022-08-07
> > > **Thank you for reading our response!**
> > >
> > > Thank you for reading our response!

---

> ### Author Response · Authors · 2022-08-01
> **Response to the Reviewer v9y3**
>
> Thank you for your thoughtful reviews and constructive suggestions!
>
> ### 1. How the proposed approach generalized to larger language models:
> Thank you very much for your suggestion on investigating our approach with larger language models. In the following, we provide experimental results to analyze this problem from two aspects.
>
> #### 1.1. How anisotropic larger language models are:
> |Model|Model Size|Training Objective|perplexity$\\downarrow$|conicity$\\downarrow$|self-similarity$\\downarrow$|
> |:-------------:|:-------------:|:-------------:|:-------------:|:-------------:|:-------------:|
> |**Vanilla Transformer**|117M|MLE|26.60|0.50|0.22|
> |||SimCTG|**26.55**|**0.47**|**0.19**|
> |||||||
> |**GPT-2-small**|117M|MLE|24.32|0.90|0.86|
> |||SimCTG|**23.82**|**0.43**|**0.18**|
> |||||||
> |**GPT-2-medium**|345M|MLE|17.26|0.75|0.63|
> |||SimCTG|**17.10**|**0.44**|**0.18**|
> |||||||
> |**GPT-2-large**|774M|MLE|16.57|0.46|0.20|
> |||SimCTG|**16.53**|**0.42**|**0.17**|
> |||||||
> |**GPT-2-xl**|1.6B|MLE|16.10|0.45|0.20|
> |||SimCTG|**16.08**|**0.43**|**0.18**|
>
> First, we evaluate the anisotropy of language models with different sizes. To this end, we conduct experiments on Wikitext-103 by varying the size of the language model from 117M (i.e. GPT-2-small) up to 1.6B (i.e. GPT-2-xl). In addition, as suggested by Reviewer a2Vh, we also include a non-pre-trained model (i.e. vanilla transformer) with the same size as GPT-2-small and the conicity metric [1] to measure the anisotropy of language models.
>
> The experimental results are presented in the Table above from which we can draw several conclusions:
>   * (1) For all models, SimCTG helps to improve the perplexity as well as the language model's isotropy.
>   * (2) With the same number of model parameters (i.e. 117M), the non-pre-trained model (i.e. vanilla transformer) does not suffer the anisotropic problem as the pre-trained GPT-2 does.
>   * (3) For pre-trained language models, as the model size increases, the anisotropic problem becomes less severe when training with the vanilla MLE objective. Specifically, when the underlying language model is large enough (i.e. GPT-2-large and GPT-2-xl), the performances of SimCTG and MLE are comparable with each other.
>
> To conclude, the anisotropy of language models relates to two factors: (i) whether the language model is pre-trained or not; and (ii) the size of the language model. Nonetheless, SimCTG always helps under all circumstances. In the camera-ready version, we will add more discussions with respect to this aspect. We leave the full-scope and rigorous investigations on the anisotropy of larger language models to our future work.

---

### Official Review · Reviewer_SXXZ · 2022-07-11

**Rating:** 5
**Confidence:** 4
**Soundness:** 3 good
**Presentation:** 4 excellent
**Contribution:** 2 fair

**Summary:**

In this work, the authors investigate the model degeneration problem in neural text generation, they show that the main reason for model degeneration is the anisotropic distribution of token representation. Hence, they use a contrastive training objective to enlarge the distance between tokens and propose a new decoding method of contrastive search to keep the coherence in the generated text. This paper is well-written and it shows its advantages compare to other methods.

**Questions:**

1. The anisotropic distribution of token embeddings is considered from a global perspective, why Figure 1 can reveal the phenomenon? I think the phenomenon can not be explained by anisotropic.
2. In Table 1, for the metric of language modeling quality, what decoding strategies are used for different models?

**Limitations:**

The method proposed by the authors prefers to generate tokens that have not appeared in the previous context, it will influence the performance of high-frequency tokens. At the same time, this method is more suitable for tasks with high degrees of freedom.

**Strengths And Weaknesses:**

Strengths:
1. This paper is well-written, methods are clearly explained.
2. The given storyline and corresponding experiments are logically consistent.
---------
Weakness:
1. The contrastive learning method has been widely used to solve the representation degeneration problem, the method is not novel.
2. The proof is not convincing enough, it is not enough to use the token cosine similarity to reveal the model degeneration problem is caused by the token representation degeneration.
3. It needs more experiments to illustrate the effectiveness of the methods, it needs to add some representative NLG experiments, e.g. machine translation.

---

> ### Author Response · Authors · 2022-08-01
> **Response to the Reviewer SXXZ**
>
> Thank you for your questions.
>
> ### Weakness 1: Lack of novelty:
> To the best of our knowledge, our work is the first effort on applying (token-level) contrastive learning approach to improve open-ended text generation models. The novelty and originality of our work are universally acknowledged by all other reviewers (Reviewer R4YS, a2Vh and v9y3).
>
> ### Weakness 2: Reason for degeneration:
> Our work is motivated by the anisotropic nature of language models. We demonstrated that the anisotropy of language models is one of the underlying factors for model degeneration. Conversely, by maintaining an isotropic distribution of token representations, the model degeneration problem can be successfully addressed with our proposed decoding method, i.e. contrastive search.
>
> ### Weakness 3: More experiments:
> Open-ended text generation by itself is a core task in the NLP community and it is different in nature with respect to other NLG tasks, such as machine translation and document summarization, that have a low degree of freedom. In this work, our approach was specifically designed for the task of open-ended text generation. We have demonstrated the effectiveness of our approach through comprehensive experiments and analysis as acknowledged by Reviewers R4YS, a2Vh and v9y3.
>
> It is interesting to investigate how well our approach performs on other NLG tasks like machine translation. We will leave it to our future work as described in our limitation section (Appendix A).
>
> ### Question 1: Definition of anisotropy:
> The anisotropic nature of language models was first investigated by [1]. The authors' original definition of anisotropic token distribution was based on token-level cosine similarity measurement [1]. In our study, we follow the same method as [1] and illustrate the language model's anisotropy from token-level measurement as demonstrated in Figure 1. Please refer to the original paper [1] for more details.
>
> ### Question 2: How the language modelling quality is evaluated:
> Decoding algorithms are not required and only human-written texts are needed for the evaluation of language modelling quality. Please refer to Lines 140-148 of our paper and [2,3,4,5] for the definition of evaluation metrics on language modelling quality.
>
> ### Limitations:
> We never limit the model to only ***"generate tokens that have not appeared in the previous context"***. Instead, the proposed contrastive search is able to generate sequences containing a reasonable amount of repetitions, that are comparable to human-written texts, for high-frequency tokens as demonstrated in Table 1.
>
> Additionally, as per our response to weakness \#3, we focus on the task of open-ended text generation which has a high degree of freedom by its nature. Accordingly, our approach was specifically designed for this task and we have demonstrated the effectiveness of our method through comprehensive experiments and analysis as acknowledged by Reviewers R4YS, a2Vh and v9y3.
>
>
> [1] - [https://aclanthology.org/D19-1006.pdf](https://aclanthology.org/D19-1006.pdf)
>
> [2] - [https://aclanthology.org/P19-1285.pdf](https://aclanthology.org/P19-1285.pdf)
>
> [3] - [https://arxiv.org/pdf/1908.04319.pdf](https://arxiv.org/pdf/1908.04319.pdf)
>
> [4] - [https://arxiv.org/pdf/1803.10049.pdf](https://arxiv.org/pdf/1803.10049.pdf)
>
> [5] - [https://openreview.net/references/pdf?id=HJ7I_nV5g](https://openreview.net/references/pdf?id=HJ7I_nV5g)

---

> > ### Comment · Reviewer_SXXZ · 2022-08-08
> > **Thanks for the authors clarification, score raised to 5**
> >
> > The response has addressed my major concerns. However, contrastive learning and its findings for NLP are not new.
> >
> > I decide to raise my rating to 5 -- borderline accept.

---

> > > ### Author Response · Authors · 2022-08-08
> > > **Thank you for reading our response!**
> > >
> > > Thank you for reading our response!

---

### Official Review · Reviewer_a2Vh · 2022-07-12

**Rating:** 8
**Confidence:** 4
**Soundness:** 4 excellent
**Presentation:** 4 excellent
**Contribution:** 3 good

**Summary:**

NLG models rely on the maximum likelihood objective during training and a related decoding strategy. In the most vanilla formulation, this combination of training objective + inference strategy results in degenerate text. One reason is the anisotropic distribution of the token embeddings learned by the underlying models. Inspired by contrastive representation learning, this paper proposes SimCTG a new approach that incorporates an additional term in the training formulation and a modified inference time strategy during decoding. The combination induces diversity (reducing degeneration) in the generated text while maintaining relevance to the input.

**Questions:**

1. The paper mentions in Line 112 that the formulation is architecture agnostic. Would the same strategy work in the case of encoder-decoder models, where the decoder might learn the isotropic embeddings while the encoder might remain more or less unaffected? Or should the objective also involve constraining the encoder representations as well?
2. Can you please elaborate on the Informativeness evaluation in human studies? The description given in the appendix is also difficult to parse.
3. How does the formulation perform on a vanilla transformer network / standard LSTM model?

**Limitations:**

The authors have addressed the limitations in the appendix however they should discuss the potential downsides of using the GPT2 model - offensive content generation. Since the decoding process involves the generation of words that are different from the word appearing before them (contrastive search), it might help to show some cases where the model diverges too much from the content and starts rewarding offensive content (or at least mention it).

Missing citation:
"Towards Transparent and Explainable Attention Models (Mohankumar et al., ACL 2020)"

One evaluation metric for checking token dissimilarity is proposed in "Towards Understanding the Geometry of Knowledge Graph Embeddings (Chandrahas et al, ACL 2018). Kindly evaluate that since self-similarity is very similar to the objective that is being tried to optimize in the SimCTG method.

**Strengths And Weaknesses:**

Strengths:
1. The paper is well-written and easy to understand.
2. Applicable to wide variety of NLG tasks.
3. Good performance on Wikitext-103, LCCC, and Dialog Daily.
4. Comprehensive strategy analysis and generation analysis (section 6).
5. Fair comparison against competitive baselines and readable code in supplementary.

Weaknesses:
1. [Very Minor] While the contributions of this work are certainly novel, the formulation is a simple variation of previously known schemes, one of which had been explored for encoder-only models. The training strategy is a variation of TaCL [19]  + Self-similarity [6]. The decoding objective is a variation of the unlikelihood objective.
2. Human Evaluation for informativeness is vague.
3. Evaluation of non-pre-trained models is missing. This would help in understanding if the performance gains are only restricted to pre-trained models or not.

---

> ### Author Response · Authors · 2022-08-01
> **Response to the Reviewer a2Vh**
>
> ### 3. The performance of SimCTG on the vanilla model:
>
> |Model| Objective|ppl$\\downarrow$|acc$\\uparrow$|conicity$\\downarrow$|self-similarity$\\downarrow$|Method|diversity$\\uparrow$|MAUVE$\\uparrow$|coherence$\\uparrow$|
> |:-------------:|:-------------:|:-----------------------:|:-----------------------:|:-----------------------:|:-----------------------:|:-----------------------:|:-----------------------:|:-----------------------:|:-----------------------:|
> |**Vanilla Transformer**|MLE|26.60|35.62|0.50|0.22|nucleus|0.89|0.81|0.541|
> |||||||contrastive|0.90|0.83|0.561|
> ||SimCTG|**26.55**|**36.03**|**0.47**|**0.19**|nucleus|0.89|0.82|0.543|
> |||||||contrastive|**0.91**|**0.85**|**0.566**|
> |||||||||||
> |**GPT-2**|MLE|24.32|39.63|0.90|0.86|nucleus|0.94|0.90|0.577|
> |||||||contrastive|0.24|0.18|0.599|
> ||SimCTG|**23.82**|**40.91**|**0.43**|**0.18**|nucleus|0.94|0.92|0.584|
> |||||||contrastive|**0.95**|**0.94**|**0.610**|
> |||||||||||
> |**Human**|-|-|-|-|-|-|0.95|1.00|0.644|
>
> Thank you for your suggestion on testing our approach on vanilla models!
>
> We have conducted experiments with the vanilla transformer model. To make a consistent comparison, we keep the parameter size of the vanilla transformer model the same as the GPT-2 model. The automatic evaluation results are presented in the above Table, where the results of GPT-2 are copied from Table 1 of our paper. To measure the isotropy of language models, in the above Table, we also report the results of the self-similarity metric (Eq. (6) of our paper) and the conicity metric from [Chandrahas et al., 2018](https://aclanthology.org/P18-1012.pdf).
>
> From the results, we can draw several conclusions:
>
> *  (1) SimCTG also helps when training vanilla transformer model as demonstrated by the improved ppl and acc results compared with the MLE baseline, revealing the clear generalization ability of our approach on both pre-trained and non-pre-trained models.
> *  (2) For the vanilla transformer model, both MLE and SimCTG perform comparably on self-similarity and conicity metrics, indicating that the vanilla transformer model does not suffer from the anisotropic problem as the pre-trained GPT-2 does.  This finding is particularly interesting and it is worth further investigation on why the pre-training procedure harms the isotropic property of the language model. We will leave it to our future work.
> *  (3) Given the isotropic nature of the non-pre-trained model, our proposed contrastive search can also be directly applied to the vanilla transformer model trained with MLE and also yields better results than the nucleus sampling method.
> *  (4) Overall, on both pre-trained and non-pre-trained models, SimCTG + contrastive search yields the best results across the board.
>
> ### 4. Limitations:
> *  (1) **Offensive content generation:** Thank you for your suggestion on the potential drawback of our approach. We will add more discussions of our approach on social impact in the camera-ready version.
> *  (2) **Missiing reference:** Thank you for pointing out the missing reference. We will add it to in our next version.
> *  (3) **Conicity metric:** Thank you for suggesting the conicity metric [(Chandrahas et al., 2018)](https://aclanthology.org/P18-1012.pdf) for measuring the isotropy of language models. We have reported its results in our answer \#3 and we certainly think it helps to further strengthen the arguments of our paper. In the camera-ready version, we will include the results of the conicity metric in our experimental tables.

---

> ### Author Response · Authors · 2022-08-01
> **Response to the Reviewer a2Vh**
>
> Thank you for your thoughtful reviews and valuable suggestions!
>
> ### 1. How to adapt SimCTG to encoder-decoder models:
> For encoder-decoder models, we should apply SimCTG on the decoder side to let the decoder learn an isotropic representation space and let the encoder remain unchanged. Accordingly, when we use contrastive search, we can simply modify the degeneration penalty in Eq. (5) to let it only consider the hidden representations of previously generated tokens by the decoder. We will leave the test of our approach on encoder-decoder models for future work.
>
> ### 2. Elaboration on the informativeness metric:
> In the following, we provide some concrete examples to illustrate the definition of the informativeness metric.
>
> Take the sentence ___"Kujira developed from a broad area of disturbed weather as a tropical depression on April 9 well removed from any landmasses. Shortly after development, Kujira"___ as the prefix text, the generated text with different informative scores are:
>
>   * **5-score example:** _"quickly intensified in its early stages, and was upgraded to a tropical storm just two days after cyclogenesis. Strengthening slowed afterwards, though the storm attained typhoon intensity on April 14. Intensification continued and late on April 15, Kujira reached its peak intensity with winds of 165 km / h ( 105 mph ) and a minimum barometric pressure of 930 mbar ( hPa ; 27.46 inHg ). Following peak intensity, Kujira would begin to track northwest and oscillate in strength, cresting an additional two times in intensity."_
>
>     **[Discussion]** We can see that a 5-score generated text is both logically as well as grammatically fluent, and contains interesting and novel content.
>
>   * **3-score example:** _"became extratropical, which caused the storm to turn to the northeast-northwest on April 10. By the next day, Kujira had weakened to tropical storm status and was located in the Gulf of Mexico at an elevation of 1,200 m ( 3,600 ft ). It made landfall near Corpus Christi, Texas, on April 11 near the mouth of the Brazos River. The cyclone moved ashore in southeastern Texas, on April 11 near the mouth of the Brazos River."_
>
>     **[Discussion]** We can see that the last two sentences in this example are partially repeated.
>
>   * **1-score example:** _"was inundated with tropical cyclones, which caused it to become a tropical depression. The storm was accompanied by a tropical cyclone named the Tropical Storm of the Year, which caused it to become a tropical depression. Kujira was one of the most severe storms to hit the United States in the past decade. The storm was accompanied by a tropical cyclone named the Tropical Storm of the Year, which caused it to become a tropical depression. The storm was accompanied by a tropical cyclone named the Tropical Storm of the Year, which caused it to become a tropical depression."_
>
>     **[Discussion]** Obviously, in the 1-score generated text, there is less useful information or novel content. And most of its content is already displayed in the prefix text.

---

### Official Review · Reviewer_R4YS · 2022-07-14

**Rating:** 5
**Confidence:** 4
**Soundness:** 2 fair
**Presentation:** 3 good
**Contribution:** 3 good

**Summary:**

This paper aims to solve the degeneration problem using a contrastive training objective and a contractive search. The proposed contrastive training objective encourages the model to learn isotropic representation for the tokens. In addition, the proposed contrastive search algorithm scores hypotheses by discriminating the candidate tokens and previous context tokens. Experiments show improved ppl and acc on language model trained with the contrastive objective and enhanced generation qualities with contrastive search.


**Questions:**

LN158: Is there any study demonstrating the effectiveness of the metric on coherence?

Table 1: Why does contrastive search work so much better for SimCTG than for MLE and Unlike?



**Strengths And Weaknesses:**

Strength: 1) The proposed methods are relatively simple and general, which can potentially be applied to any text generation model. 2) The results strongly support the methods. 3) Well written.

Weakness: 1) The proposed methods - contrastive training objective and contrastive search - are two independent methods that have little inner connection on both the intuition and the algorithm. 2) The justification for isotropic representation and contractive search could be more solid.

---

> ### Author Response · Authors · 2022-08-01
> **Response to the Reviewer R4YS**
>
> Thank you for your valuable suggestions and questions!
>
> ### 1. Inner connection between SimCTG and contrastive search:
> Actually, we can draw a nice connection between SimCTG and contrastive search.
> * (1) The goal of SimCTG is to let the language model learn an isotropic representation space from human-written texts. To put it in another way, given a **human-written text**, SimCTG encourages the language model to obtain an isotropic and discriminative token similarity matrix in which only the diagonal entries have high similarities as shown in Figure 1(b).
> * (2) On the other hand, as mentioned in Lines 36-37, contrastive search aims to keep the isotropic property (i.e., spareness) of the token similarity matrix of the **generated text**. Therefore, the text generated by contrastive search is more similar to human-written text.
>
> To conclude, SimCTG enables the language model to obtain isotropic representations with human-written texts, while contrastive search maintains the isotropic property of the text generated by the language model.
>
> **[Visual Demonstration]** We also provide a visual demonstration of the connection between SimCTG and contrastive search at Figure 3 in Appendix I. From which we see that, only by combining SimCTG and contrastive search, we can obatin a nice and isotropic token similarity matrix for both the human-written text (i.e., the prefix text) as shown in the red box and the generated text as shown in the yellow box.
>
> ### 2. Justification for isotropic representation and contrastive search could be more solid:
> Thank you for your suggestion. In the camera-ready version, we will include the details from our answer \#1 to further emphasize the connection between isotropic representation and contrastive search.
>
> ### 3. Effectiveness of the metric on coherence:
> We have considered all the baseline automatic metrics we could find in the recent literature. However, for open-ended text generation tasks, we are not able to find an existing metric that automatically measures the semantic coherence between the prefix text and the generated text. To this end, we propose to use a strong sentence embedding method, SimCSE, to automatically measure the coherence between the prefix text and the generated text. More importantly, we conduct extensive human evaluations to further assure the advantage of our approach in terms of the coherence aspect. The human evaluation results validate that our method indeed generates significantly more coherent text compared to other baseline methods, which is in line with the results acquired by our proposed coherence metric.
>
> ### 4. Why contrastive search works much better on SimCTG:
> The reason why contrastive search works best on SimCTG is that MLE and unlikelihood cannot obtain an isotropic representation space. As we describe in Lines 186-187, when the language model's representation space is anisotropic, the degeneration penalty $\\max\\{s(h_v, h_{x_j}):1\leq j \leq t-1\\}$ in Eq. (5) of different token candidates $v$ become indistinguishable with respect to each other, making contrastive search less effective.
>
> Let's consider a simple example. Suppose the representation space of the language model is extremely anisotropic such that the representations of all tokens are identical. Therefore, the cosine similarity between representations of any two tokens is always 1.0. In this case, when we apply contrastive search, the degeneration penalty for all candidate tokens would all be 1.0, i.e. all identical. Therefore, the selection of the output token will only depend on the model confidence term in Eq. (5), making contrastive search degenerate to the vanilla greedy search which further leads to unsatisfactory performance.
>
> As we demonstrate in Section 6.1, the representation space of SimCTG is much more isotropic than MLE and Unlikelihood. As a result, contrastive search works best on SimCTG.

---

### Author Response · Authors · 2022-08-01
**Thank you very much for your thorough reviews!**

Thank you for the comprehensive reviews and thoughtful comments. We are delighted that reviewers appreciated the novelty and originality of the paper.

We are excited with the recognition that "The proposed methods are relatively simple and general" and that our approach is "Applicable to wide variety of NLG tasks". We are thrilled with the reviewers' acknowledgment that "The main experiments are complemented with good analysis experiments" and that "Comprehensive strategy analysis and generation analysis". We are also pleased that the reviewers found that "This paper has good originality, clarity, quality" and "The paper is well-written and easy to understand".

Below, we respond to each reviewer separately. Please let us know if you have additional questions or comments!

---

### Meta-Review · Area_Chair_qef8 · 2022-08-25

**Recommendation:** Accept
**Confidence:** Certain

**Metareview:**

All four reviewers sided to accept the paper, as the proposed contrastive search approach to mitigating text degeneration problem is simple and effective and has applications to a variety of NLG tasks. Its evaluation is quite comprehensive and includes competitive baselines, human evaluation, and evaluation of both LM/generation quality on Wikitext-103 and effect on a downstream task (dialog). Two of the reviewers were more hesitant (borderline accept), but one of them was quite satisfied with the author response and the other reviewer didn't raise any major issue. The one remaining concern is that experiments with GPT-2 were base on the "small" model, but the rebuttal shows that the findings of the paper mostly hold with bigger language models (medium and large) but become relatively small with XL. We suggest including these additional experiments in the next version of the paper, along with further discussions of these smaller differences.

**Award:**

No

---

### Decision · Program_Chairs · 2022-09-14

Accept